# The landscape of antibody binding affinity in SARS-CoV-2 Omicron BA.1 evolution

Alief Moulana[1†], Thomas Dupic[1†], Angela M Phillips[1*], Jeffrey Chang[1,2†], Anne A Roffler[3], Allison J Greaney[4,5,6], Tyler N Starr[4], Jesse D Bloom[4,5,7], Michael M Desai[1,2,8,9*]

[1]Department of Organismic and Evolutionary Biology, Harvard University, Cambridge, United States; [2]Department of Physics, Harvard University, Cambridge, United States; [3]Biological and Biomedical Sciences, Harvard Medical School, Boston, United States; [4]Basic Sciences Division and Computational Biology Program, Fred Hutchinson Cancer Research Center, Seattle, United States; [5]Department of Genome Sciences, University of Washington, Seattle, United States; [6]Medical Scientist Training Program, University of Washington, Seattle, United States; [7]Howard Hughes Medical Institute, Seattle, United States; [8]NSF-Simons Center for Mathematical and Statistical Analysis of Biology, Harvard University, Cambridge, United States; [9]Quantitative Biology Initiative, Harvard University, Cambridge, United States

**\*For correspondence:**
angela.phillips@ucsf.edu (AMP);
mdesai@oeb.harvard.edu (MMD)

[†]These authors contributed equally to this work

**Abstract** The Omicron BA.1 variant of SARS-CoV-2 escapes convalescent sera and monoclonal antibodies that are effective against earlier strains of the virus. This immune evasion is largely a consequence of mutations in the BA.1 receptor binding domain (RBD), the major antigenic target of SARS-CoV-2. Previous studies have identified several key RBD mutations leading to escape from most antibodies. However, little is known about how these escape mutations interact with each other and with other mutations in the RBD. Here, we systematically map these interactions by measuring the binding affinity of all possible combinations of these 15 RBD mutations ($2^{15}$=32,768 genotypes) to 4 monoclonal antibodies (LY-CoV016, LY-CoV555, REGN10987, and S309) with distinct epitopes. We find that BA.1 can lose affinity to diverse antibodies by acquiring a few large-effect mutations and can reduce affinity to others through several small-effect mutations. However, our results also reveal alternative pathways to antibody escape that does not include every large-effect mutation. Moreover, epistatic interactions are shown to constrain affinity decline in S309 but only modestly shape the affinity landscapes of other antibodies. Together with previous work on the ACE2 affinity landscape, our results suggest that the escape of each antibody is mediated by distinct groups of mutations, whose deleterious effects on ACE2 affinity are compensated by another distinct group of mutations (most notably Q498R and N501Y).

## Editor's evaluation

This fundamental study, dealing with antibody binding to spike protein of the omicron variant of SARS CoV2, advances our insights into antibody escape and the importance of epistasis in antibody binding. The evidence is rigorous and compelling. The work will be of great interest to investigators in the field of evolutionary biology/medicine, immunologists, and virologists.

## Introduction

In November 2021, the SARS-CoV-2 Omicron BA.1 variant emerged and quickly rose to high frequency worldwide, in part due to its ability to escape preexisting immunity (*Cao et al., 2022*; *Ao et al., 2022*; *Planas et al., 2022*; *Viana et al., 2022*). This immune escape is mediated by mutations in the receptor binding domain (RBD) of the spike protein, which is the major target of SARS-CoV-2 neutralizing antibodies (*Greaney et al., 2021b*; *Greaney et al., 2021c*; *Iketani et al., 2022*; *Dai and Gao, 2021*). Antibodies targeting the RBD can bind different epitopes, and they have been grouped into several classes (*Barnes et al., 2020b*; *Barnes et al., 2020a*). Some previous SARS-CoV-2 variants which have a subset of the 15 mutations found in the BA.1 RBD (e.g. K417N, N501Y in Beta and T478K, N501Y in Delta) can evade some antibodies of certain epitope classes but still bind to others (*Liu et al., 2021*; *Zhou et al., 2021*; *Greaney et al., 2022*; *Greaney et al., 2021a*). In contrast, BA.1 can escape most antibodies that bind to very distinct epitopes, including antibodies elicited by previously circulating variants (*Cao et al., 2022*; *Dejnirattisai et al., 2022*; *Cameroni et al., 2022*).

Existing studies of SARS-CoV-2 immune escape have focused on measuring the effects of single mutations (or, in some cases, of a small subset of mutations) on antibody escape in the context of specific SARS-CoV-2 variants (*Dejnirattisai et al., 2022*; *Starr et al., 2021b*; *Starr et al., 2021a*). However, simultaneous escape of most antibodies is likely to require multiple mutations, and it is unclear how these mutations might interact. A large body of work has demonstrated that the specific combination of mutations in the BA.1 variant can evade various antibodies of distinct epitopes (*Cao et al., 2022*; *Planas et al., 2022*; *Dejnirattisai et al., 2022*; *Chakraborty et al., 2022*). However, the landscape on which this evolution occurred is not well understood. Do mutations involved in escape from one antibody with a certain epitope interfere with the effects of those involved in escaping others with different contact sites, or are the effects largely independent? And how are these effects mediated by epistatic interactions with other mutations in the RBD?

As we observed in previous work, several of these antibody-escape mutations also reduce affinity to ACE2, suggesting that they were positively selected because they contribute to immune escape (*Starr et al., 2020*; *Mannar et al., 2022*; *McCallum et al., 2022*). Importantly, epistatic interactions between these mutations dramatically impact ACE2 affinity and may also differentially impact the escape of antibodies with very different epitopes (*Moulana et al., 2022*; *Starr et al., 2022a*). For example, escape from some antibodies like S309 has been difficult to attribute to specific mutations (*McCallum et al., 2022*; *Case et al., 2022*), perhaps because measurements have so far been limited to single mutations. These observations suggest that we need to more comprehensively characterize the role of epistasis and potential trade-offs to understand the simultaneous evolution of escape from multiple antibodies of distinct epitopes and ACE2 binding affinity.

Here, to understand how immune pressure may have shaped the evolution of BA.1, we measured the equilibrium binding affinities ($K_{D, app}$) of the spike protein RBD to four therapeutic monoclonal antibodies (mAbs) with distinct RBD epitopes: LY-CoV016, LY-CoV555, REGN10987, and S309, for all possible evolutionary intermediates between the ancestral Wuhan Hu-1 RBD and the BA.1 variant. This set of antibodies includes the primary epitopes generally covered by therapeutic mAbs (*Barnes et al., 2020a*; *Cameroni et al., 2022*). The first three antibodies are fully escaped by Omicron BA.1, while S309 has reduced affinity. We find that for each antibody, only a few mutations significantly impact affinity, and these mutations are largely (but not entirely) orthogonal between the four antibodies. Additionally, we find that epistasis plays a limited role in determining affinity to antibodies that are fully escaped by BA.1 but contributes substantially to the reduced affinity for the partially escaped antibody, S309. Together, this work systematically characterizes how SARS-CoV-2 can evade distinct RBD-targeted antibodies while maintaining ACE2 affinity.

## Results

In previous work, we generated a combinatorially complete library comprising all possible intermediates between the ancestral SARS-CoV-2 Wuhan Hu-1 spike protein RBD and the Omicron BA.1 variant (*Moulana et al., 2022*). The BA.1 RBD differs from Wuhan-1 by 15 amino acid substitutions, so this library contains 2 (*Dejnirattisai et al., 2022*) variants containing all possible combinations of these 15 mutations. This RBD library is displayed on the surface of yeast, such that each yeast cell expresses a single variant. Here, we use Tite-seq (a high-throughput method that integrates flow cytometry and

sequencing *Moulana et al., 2022*; *Adams et al., 2016*; *Phillips et al., 2021*; *Starr et al., 2022b*; see *Figure 1—figure supplement 1A*) to measure the equilibrium binding affinities of all 32,768 variants to four antibodies with different epitopes (LY-CoV016, LY-CoV555, REGN10987, and S309). The resulting $K_{D, app}$ correlates between biological duplicates and with isogenic measurements made by flow cytometry (*Figure 1—figure supplement 1A*, *Supplementary file 1*).

Of the 32,768 variants in our library, we obtain $K_{D, app}$ for at least ~30,000 variants to each of the mAbs (32,603 for LY-CoV016, 31479 for REGN10987, 27485 for LY-CoV555, and 32650 for S309) after removing variants with poor titration curves ($r^2 < 0.8$ or $\sigma > 1$; see Methods). These $K_{D,app}$ range from 0.1 nM to 1 µM (which is our limit of detection and likely corresponds to non-specific binding), with 51% of the variants fully escaping LY-CoV016 (defined as having $K_{D,app}$ above the limit of detection), 65% fully escaping LY-CoV555, 36% fully escaping REGN10897, and no variants fully escaping S309 (*Figure 1A*; see https://desai-lab.github.io/wuhan_to_omicron/ (*Johnson and Dupic, 2022*) for an interactive data browser). Escape from LY-CoV016, LY-CoV555, and REGN10897 is mediated by one or a few strong-effect mutations, with other mutations more subtly impacting affinity (*Figure 1B*). In general, strong-effect mutations make substantial contact with the corresponding antibody. Consistent with previous studies (*Greaney et al., 2021a*; *Cameroni et al., 2022*; *Starr et al., 2021b*; *Windsor et al., 2022*), these strong-effect mutations are largely distinct for each antibody, which presumably reflects their non-overlapping footprints on the RBD (*Figure 1C*) and suggests that evolution of escape from each antibody can be, to some extent, orthogonal.

The picture is more complex for S309, where BA.1 has reduced affinity relative to Wuhan Hu-1, but ~17% of variants have lower affinity than BA.1. These differences are not attributable to one or two strong effect mutations (*Figure 1A–B*). In addition, although most mutations reduce affinity, three mutations have small positive effects (on average across all backgrounds at the other loci): S375F for LY-CoV016 and E484A and N501Y for REGN10987 (*Figure 1B*). Intriguingly, each of these mutations reduces affinity to at least one of the other antibodies, and N501Y significantly improves binding to ACE2, suggesting a potential role for trade-offs (and/or epistasis that mitigates these effects on specific backgrounds).

For each antibody, binding affinities generally decrease as the number of mutations increase (*Figure 1D–G*). For LY-CoV016, LY-CoV555, and REGN10897, this trend is observed amongst variants with and without the large-effect escape mutations (*Figure 1D–F*). For LY-CoV016, K417N is sufficient for escape (*Figure 1D*, green), whereas both LY-CoV555 and REGN10987 require at least two mutations for complete escape. For LY-CoV555, both E484A and Q493R decrease affinity drastically (1000- and 100-fold, respectively), but only the combination of both mutations lead to complete escape. Complete escape from REGN10987 also requires two mutations (N440K and G446S), but the individual effects of these mutations are more subtle (reducing affinity by 5- and 10-fold, respectively). For S309, affinity declines after a few mutations and in some backgrounds increases upon further mutation, suggesting that interactions between these mutations are important in determining affinity (*Figure 1G*).

## Mostly orthogonal large-effect mutations

We first focused on analyzing how mutations and combinations of mutations lead to complete escape (defined as $K_{D,app}$ above our limit of detection) for specific sets of antibodies. To do so, we analyze the enrichment of specific mutations among non-binders (*Figure 2A*). We find a largely orthogonal set of one or two mutations are enriched among variants that do not bind each antibody: almost all variants that do not bind LY-CoV016 contain K417N, almost all variants that do not bind REGN10987 contain G446S, and many also contain N440K, and E484A and Q493R are highly enriched among variants that do not bind LY-CoV555. These escape mutations were already identified on the Wuhan background (*Tada et al., 2022*; *Starr et al., 2021c*; *Zhou et al., 2022*). The fact that different sites are involved for each antibody suggests that the RBD can evolve to independently escape antibodies with each distinct epitope, and mutations can to some extent act independently on binding to each antibody.

To analyze this further, we calculate the percentage of genetic backgrounds on which each mutation leads to complete escape from a specific antibody (i.e. that mutation converts a variant with measurable $K_{D,app}$ to a $K_{D,app}$ above our limit of detection). We see that for each antibody, one or two mutations abrogate binding. These sets of mutations are largely orthogonal among antibodies (*Figure 2B*), consistent with the enrichment analysis (*Figure 2A*). Specifically, K417N always abrogates

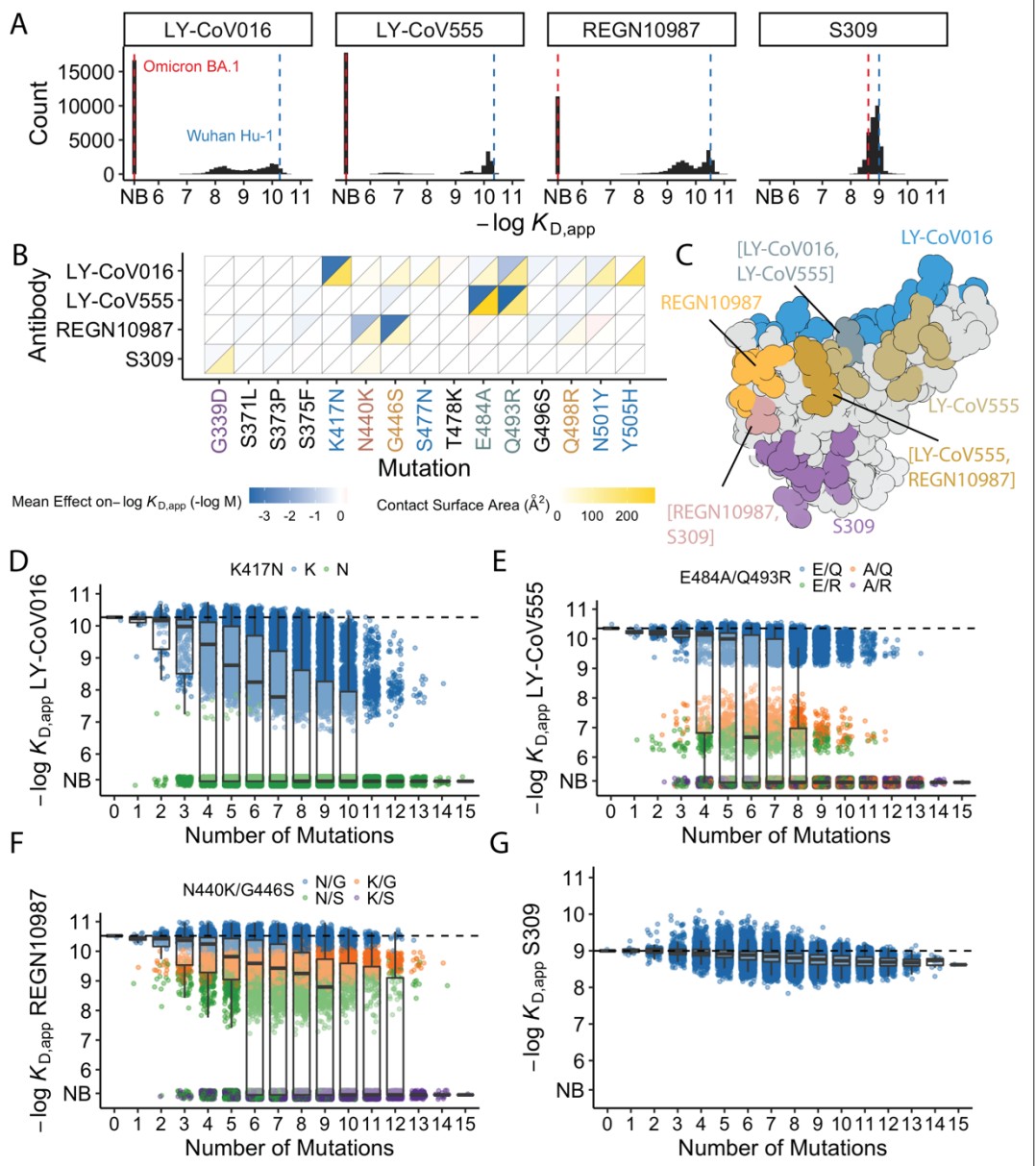

**Figure 1.** Antibody affinity landscape. (**A**) Binding affinities to one antibody from each class (LY-CoV016, LY-CoV555, REGN10987, and S309, from classes 1–4, respectively) across all N=32,768 receptor binding domain (RBD) genotypes tested. Binding affinities are shown as $-\log K_{D,app}$; vertical blue and red dashed lines indicate the $-\log K_{D,app}$ for Wuhan Hu-1 and Omicron BA.1, respectively. 'NB' denotes non-binding (escape). (**B**) Mean effect of each mutation on antibody and ACE2 affinity (defined as the change in $-\log K_{D,app}$ resulting from mutation averaged across all backgrounds at the other loci) plotted with contact surface area between each residue and each antibody. Mutations are colored by footprint highlighted in (**C**). (**C**) Structure of SARS-CoV-2 BA.1 RBD with each antibody footprint annotated (PDB ID 7 KMG, 6WPT, 7C01, and 6XDG). Residues with overlapping footprints are colored and labeled accordingly. (**D–G**) Distribution of binding affinities to different antibodies grouped by number of Omicron BA.1 mutations. Binding affinity of the Wuhan Hu-1 variant is indicated by horizontal dashed lines. Variants with antibody escape mutations of interest are colored as noted in each key. NB denotes non-binding (escape). In all figures, the boxplots boxes show the spread between the 25th and 75th percentiles, with the median indicated by a horizontal line.

The online version of this article includes the following figure supplement(s) for figure 1:

**Figure supplement 1.** Schematic overview of the Tite-seq method and reproducibility of dissociation constants.

**Figure supplement 2.** Distribution of maximum log-fluorescence difference.

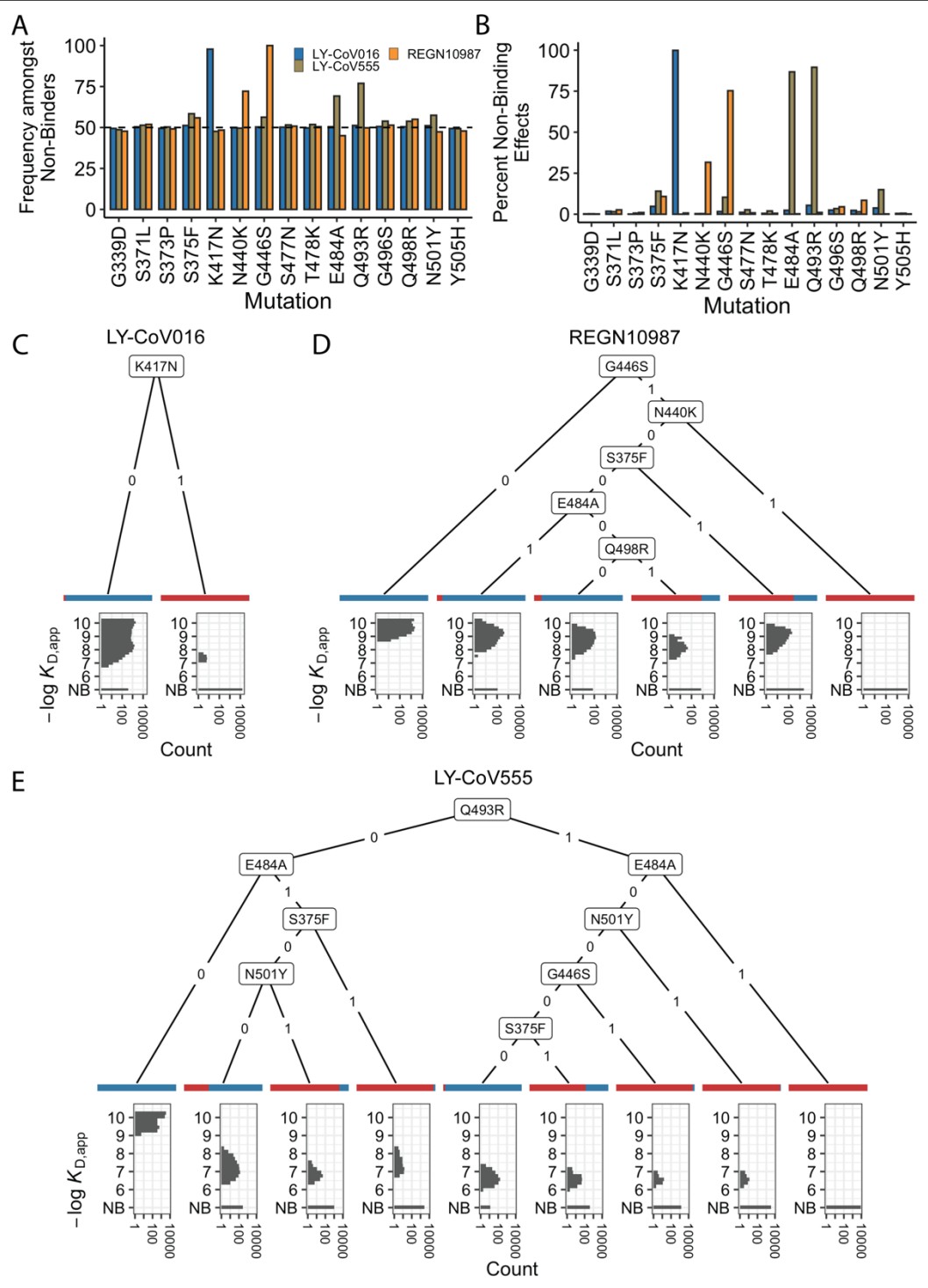

**Figure 2.** Escape mutations and genotypes. (**A**) Fraction of antibody-escaping genotypes with each mutation. (**B**) Fraction of variants for which a given mutation confers antibody escape. Effects are colored as in (**A**). (**C–E**) Decision trees of escape phenotype for each antibody modeled as a function of the mutations present. Each leaf is annotated by the proportion of the genotypes that escape the corresponding antibody (red: escape and blue: does not) and by corresponding affinity distribution. NB denotes non-binding.

binding to LY-CoV016, G446S and N440K often abrogate binding to REGN10987, and E484A and Q493R often abrogate binding to LY-CoV555.

However, we note that this orthogonality is not complete. For example, G446S is slightly enriched among LY-CoV555 binders, while mutation E484A is slightly depleted among variants that do not bind REGN10987 (*Figure 2A*). Consistent with this, G446S sometimes abrogates binding to LY-CoV555 (*Figure 2B*). In addition, some apparently smaller-effect mutations can be involved in abolishing binding to multiple antibodies. For example, S375F is weakly enriched among variants that do not bind REGN10987 and LY-CoV555 and often abrogates binding to these two antibodies, with G496S, Q498R, and N501Y also playing a role.

To summarize how these different mutations can act individually or in combination to lead to antibody escape, we inferred a decision tree to classify variants as binders or non-binders. To do so, for each antibody, we calculate the mutation that maximally partitions the variants into binders or non-binders. If this partitioning is not perfect, we then calculate the second mutation that maximally partitions the variants conditional on each possible state of the first site. We then proceed to further partition variants based on additional mutations in the same way until the variants are perfectly partitioned or no further mutations can significantly improve the partitioning (see Methods). We show the corresponding decision trees for LY-CoV016, REGN10987, and LY-CoV555 in *Figure 2C, D and E*, respectively. As expected, the tree associated with LY-CoV016 is very simple: the mutation K417N perfectly partitions the variants into binders and non-binders. In contrast, the trees for REGN10987 and LY-CoV555 have more complex structures, reflecting the fact that it is possible to abrogate affinity to these antibodies via multiple distinct combinations of mutations. For example, variants can escape REGN10987 by acquiring G446S and N440K (100%), or alternatively, with S375F and G446S (89%, as additional mutations may also be required). For LY-CoV555, different sets of mutations can lead to escape (e.g. Q493R and G446S or E484A and S375F). Some of these mutations partially overlap with those for REGN10987 (i.e. they are not fully orthogonal), suggesting that selection pressure from one antibody could promote subsequent escape of another.

## Inference of epistatic affinity landscapes

In addition to large-effect mutations which lead to complete escape of specific antibodies, a variety of other sites contribute to more subtle but potentially important changes in binding affinities. To analyze these subtle effects as well as the large-effect mutations leading to escape, we defined a linear model for $-\log(K_{D, app})$ as the sum of single (additive) mutational effects plus interaction terms up to a specified order (note that because $-\log[K_{D, app}]$ is proportional to the free energy of binding, we expect it to behave additively in the absence of epistatic interactions). Because non-binding variants have $-\log(K_{D, app})$ beyond our limit of detection, we fit a Tobit model (a class of regression model capable of handling truncated measurements; see Methods for details) using maximum likelihood with an L2-norm Lasso regularization. Specifically, we partition our data into training (90%) and test (10%) sets and use the training dataset to fit epistatic coefficients to a linear model truncated at each order (e.g. truncating to first-order yields additive mutational effects, second-order includes both additive effects and pairwise terms, etc.). We then evaluate performance (as the coefficient of variation) of each model on the held-out test dataset and compare the model performance using $-\log(K_{D, app})$ for each of the antibodies and ACE2 (*Figure 3A*).

We find that adding epistatic interactions improves the predictive power of the model for all four antibodies as well as for binding to ACE2, though the optimal order varies (*Figure 3A*). This indicates that epistasis does play a significant role in all cases (up to second order for REGN10987, to third order for LY-CoV555, and to fourth or higher order for LY-CoV016, S309, and ACE2). The additive, pairwise, and higher-order coefficients resulting from these models are summarized in *Figure 3— figure supplement 1*. In general, we find many strong interactions across several positions in each antibody, involving both the sites that strongly determine escape variants for that antibody (e.g. between N440K and G446S for REGN10987) as well as others.

Notably, we find that the higher-order epistasis plays a much stronger role in determining affinity for ACE2 than for the three antibodies fully escaped by BA.1 (*Figure 3B*). This reflects the impact of a few strong-effect mutations in determining affinity for LY-CoV555, LY-CoV016, and REGN10987 and the role of compensatory epistasis in determining ACE2 affinity. In other words, while epistasis

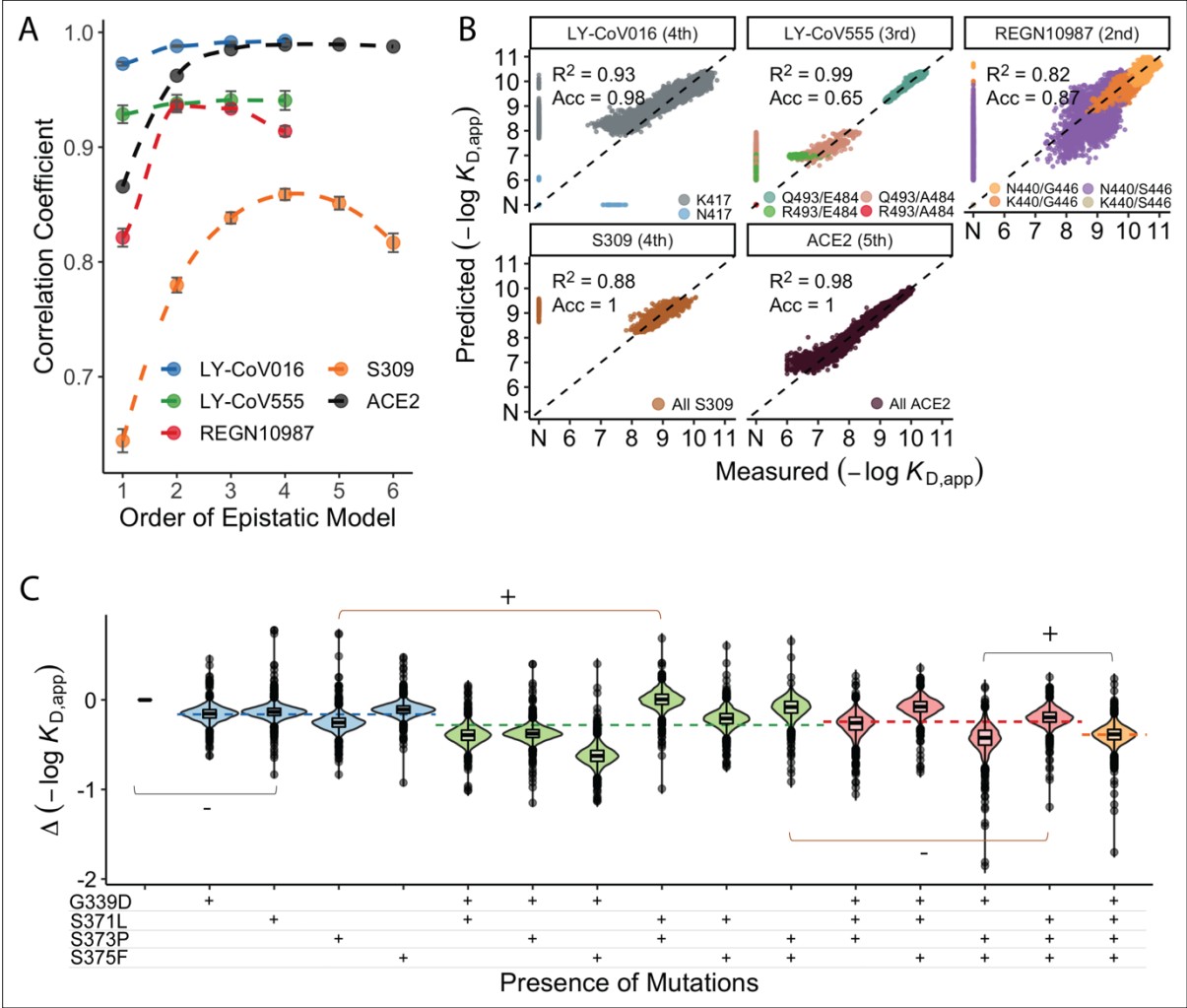

**Figure 3.** Epistatic effects on antibody binding. (**A**) Correlation coefficients between the measured values of $-\log(K_{D,app})$ and the model estimate for various orders of epistatic models. Correlations are computed on the hold-out subset averaged over 10-folds of cross-validation. Zoomed-in version for orders 3–6 (sample size n=10) (**B**) Binding affinities predicted by complete coefficients from the optimum epistasis model are compared to the measured binding affinities for each antibody. Points are colored by mutations present in the genotypes, 'N' corresponds to non-binding genotypes. The accuracy measures the quality of the binary classification between binders and non-binders, and the coefficient of determination $R^2$ refers to the correlation between inferred and measured binding affinities, excluding non-binders. (**C**) Effects of mutations G339D, S371L, S373P, and S375F on S309 affinity grouped by the presence of each mutation. Each violin color corresponds to the number of mutations considered. Dashed line color denotes the average effect for each group represented by the violin color. The gray and orange lines indicate cases where the addition of mutation S371L has a positive (+) or negative (−) effect depending on the background (sample size n=2048).

The online version of this article includes the following figure supplement(s) for figure 3:

**Figure supplement 1.** Epistatic coefficients.

is relevant for all measured phenotypes, antibody escape is more simply determined by the additive effects of individual mutations, while maintaining ACE2 affinity involves more complex epistatic interactions.

High-order epistatic interactions are also important in determining affinity to S309. In *Figure 3C*, we highlight four neighboring mutations (G339D, S371L, S373P, and S375F) which interact non-additively to produce the reduction in affinity observed in BA.1 relative to Wuhan Hu-1. Each of these mutations weakly reduces affinity on their own, and specific combinations of these mutations can reduce affinity by up to two orders of magnitude, but the reduction in affinity resulting from all four mutations is less than some sets of three mutations. These patterns emerge from a complex set of high-order epistatic interactions among the mutations. For example, S371L reduces affinity on the Wuhan Hu-1 background but increases affinity on the background containing G339D, S373P, and S375F (and without

G339D, S371L increases affinity in the presence of S373P if S375F is absent but not if it is present). Thus, some variants lacking S371L evade S309 more effectively than BA.1, and interestingly we note that this mutation is absent in BA.2 and BA.3 and replaced instead by S371F.

## Tradeoffs between antibody and ACE2 affinities

In previous work, we found that antibody escape mutations (as defined in earlier studies) typically reduce ACE2 affinity, suggesting that viral evolution is constrained by a tradeoff between immune evasion and the ability to enter host cells. Consistent with this, we find here that variants that escape one or more antibodies (as defined by the data reported in this work) but have few additional mutations have reduced ACE2 affinity relative to Wuhan Hu-1. However, as additional BA.1 mutations are accumulated, the ACE2 binding affinity tends to increase until it exceeds the Wuhan Hu-1 value even in the presence of multiple antibody escape mutations (*Figure 4A*). This suggests that the evolution of the BA.1 variant is driven both by immune escape and the need for compensatory mutations that mitigate the negative effects of the escape mutations on ACE2 binding.

The strength of this tradeoff and the potential importance of compensatory evolution are distinct between the different antibodies (*Figure 4B and C*). For example, escape from LY-CoV016 or LY-CoV555 reduces ACE2 binding affinity in the absence of compensatory mutations (Q498R and N501Y) but not in their presence (*Figure 4B*). In contrast, REGN10987 escape does not strongly reduce affinity to ACE2, whether or not Q498R and N501Y are present. However, this tradeoff is likely relevant overall, as escaping all three antibodies substantially reduces ACE2 affinity in the absence of Q498R and N501Y, while the reduction in ACE2 affinity is minimal in their presence. Consistent with this general picture, the frequency of most escape mutations (G446S, E484A, and Q493R) is higher across the SARS-CoV-2 phylogeny in the presence of compensatory mutations (*Figure 4D*), though we note that this is not universally true (e.g. the frequency of N440K is only slightly higher in the presence of compensatory mutations, and the frequency of K417N is lower with the compensatory mutations).

Although antibody escape mutations do tend to reduce ACE2 affinity, antibody binding affinity (but not complete escape) is not strongly correlated with ACE2 affinity (*Figure 4C*). The details of this relationship vary by antibody. For LY-CoV016 and LY–CoV555, there is a weak overall positive correlation (i.e. lower antibody affinity also tends to correspond to reduced ACE2 affinity). However, this correlation is dominated by the variants that lack the compensatory mutations at sites 498 and 501; in the presence of Q498R and N501Y, the correlation is reduced, especially for LY-CoV016. For REGN10987 and S309, there is a similar weak overall positive correlation, which is not dependent on Q498R and N501Y. We also note that while compensatory mutations Q498R and N501Y largely drive the variance in ACE2 affinity, they minimally impact antibody binding affinities (*Figure 4D*).

## Discussion

Overall, we find that BA.1 escape from LY-CoV016, LY-CoV555, and REGN10987 is driven by a relatively small set of mutations: K417N for LY-CoV016, N440K and G446S for REGN10987, and E484A and Q493R for LY-CoV555. These mutations have largely orthogonal effects on affinity to the three antibodies, suggesting that the evolution of escape to each can occur independently, as might be expected given the distinct epitopes they target (*Barnes et al., 2020b*; *Cameroni et al., 2022*; *Starr et al., 2021a*).

However, despite these largely orthogonal effects of large-effect mutations on antibody escape, we do observe limited trade-offs among LY-CoV016, LY-CoV555, and REGN10987, with four mutations (S375F, T478K, E484A, and N501Y) improving affinity to one antibody and reducing affinity to another. These positive effects are modest compared to the reductions in affinity caused by other mutations. In fact, for all antibodies studied here, outside of a few large-effect mutations that abrogate or nearly abrogate binding, most mutations weakly impact binding affinity and, even collectively, are insufficient to abrogate it.

In contrast to the orthogonality of antibody escape, trade-offs between binding ACE2 and escaping antibodies are much stronger. While the mutations with a small effect on antibody escape are mostly uncorrelated to ACE2 affinity, the strong-effect mutations substantially reduce ACE2 affinity. Thus, ACE2 affinity is lower for variants that escape a larger number of antibodies unless compensatory

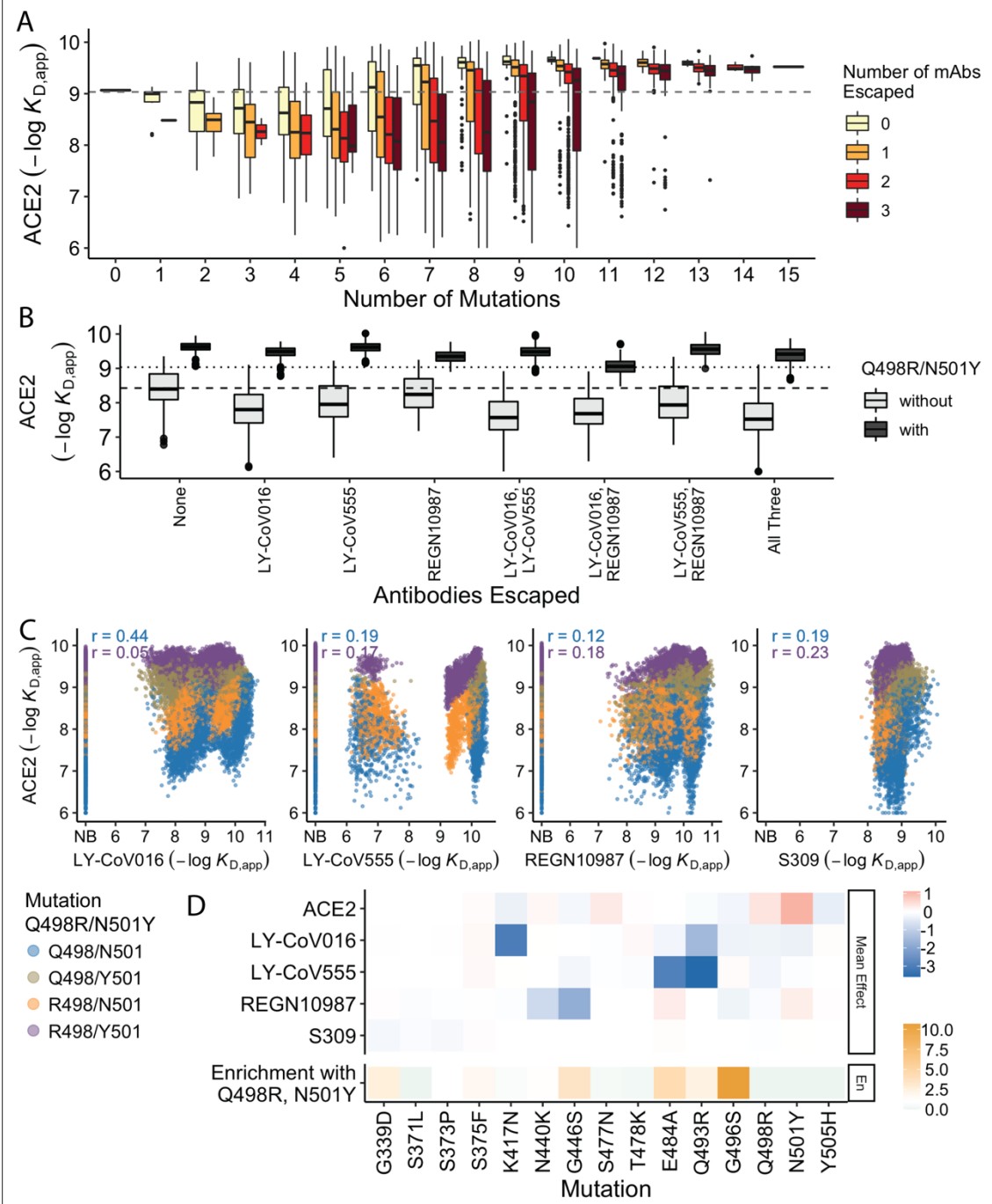

**Figure 4.** Trade-offs and comparison with ACE2 affinity. (**A**) Distribution of ACE2 binding affinity grouped by number of BA.1 mutations and the number of monoclonal antibodies escaped. The dashed line corresponds to the affinity of the Wuhan strain. (**B**) ACE2 affinity distribution grouped by antibodies escaped and the presence of compensatory mutations (N501Y and Q498R). The dotted line represents the affinity of the Wuhan strain, while the dashed line shows the average affinity of the genotypes without compensatory mutations that bind all antibodies. (**C**) Affinities to monoclonal antibodies plotted as a function of the ACE2 affinity for all genotypes. Points are colored by presence of Q498R and N501Y. (**D**) Mean effect (averaged over all backgrounds at the other loci) of each mutation on antibody affinity and on ACE2 affinity (red-blue colormap) compared to the enrichment of their frequency with Q498R and N501Y (orange colormap). The enrichment score is defined as the normalized frequency a mutation emerged on a branch on which mutations Q498R and N501Y appear divided by the normalized frequency it emerged on any intermediate background between Wuhan and BA.1.

mutations are acquired, suggesting that these compensatory mutations potentiated the establishment of the antibody escape mutations (*Moulana et al., 2022*; *Javanmardi et al., 2022*).

We also find that epistatic interactions are important in determining antibody affinity. This is particularly true for S309. Prior to this work, the reduced affinity of S309 to Omicron could not be attributed to specific mutations (*Cameroni et al., 2022*; *Case et al., 2022*). Here, we find that this ambiguity can be resolved by examining higher-order interactions between mutations, as the reduction in affinity is attributable to a fourth-order epistatic interaction. This finding suggests that the potential for future SARS-CoV-2 lineages to escape S309 and similar antibodies could depend on epistatic interactions between emerging mutations.

We note that our study focuses only on binding affinities, which may not always perfectly reflect viral escape from antibody neutralization (*Steckbeck et al., 2005*; *Culp et al., 2007*). In particular, some of the binding affinities we measure could be too weak to be physiologically relevant, and mutations may impact neutralization without impacting binding affinity significantly. However, because neutralization cannot occur in the absence of affinity, our measurements are likely to be relevant for understanding the reduced sensitivity of BA.1 to these antibodies. We also note that practical constraints limit us to studying four antibodies. This limits the generalizability of our results, particularly in light of previous structural studies which have revealed more epitopes bound by mAbs.

In spite of these limitations, our binding affinity landscapes reveal that BA.1 can escape diverse antibodies by acquiring a few large-effect mutations and can reduce affinity to others by accumulating several small-effect mutations. For the first three antibodies, one or two mutations are sufficient for total escape. However, in some cases, additional mutations can restore affinity, and in others, specific combinations of large- and small-effect mutations can abrogate affinity. Thus, despite the seemingly simple landscape of antibody escape, there are alternative, more intricate pathways that can abrogate affinity. In contrast, for the S309 antibody, four mutations drive the decline in affinity yet are also involved in higher-order epistatic interactions that counteract this decline. This epistasis results in an affinity threshold, beyond which additional mutations do not reduce affinity.

More generally, this work illustrates a broad diversity of ways in which epistasis can shape a protein landscape. While our earlier work demonstrates that the RBD ACE2 affinity landscape is defined by several epistatic interactions (*Moulana et al., 2022*), our results here show that escape from antibody binding can in many (but not all) cases be driven by individual mutations. It is unclear why these landscapes involve such distinct patterns of epistasis. One possibility is that the prevalence of epistasis in the ACE2 affinity landscape is a consequence of the long history of spike protein evolution to bind ACE2, which may have selected for more complex epistatic interactions among the acquired mutations over time (in contrast, evolution to escape antibodies is relatively recent). Alternatively, the relative simplicity of the antibody escape landscapes could be attributed to the fact that beneficial mutations in this case disrupt (rather than maintain or improve) an interaction (*Greaney et al., 2021a*; *Starr et al., 2021b*; *Starr et al., 2020*). Of course, these possibilities are not mutually exclusive (*Rotem et al., 2018*), and discriminating between them will require further investigation into other viral protein and antibody landscapes to determine how broadly the patterns we observe in this study generalize to other systems.

Predicting the future evolution of the Omicron lineage will also require determining how these affinity landscapes translate to immune evasion, how antibody affinity landscapes vary within a class or epitope group, and how mutations beyond this set may further enhance immune evasion. For example, neutralization assays with minimally mutated genotypes would confirm whether the strong-effect mutations are indeed sufficient for escape. Furthermore, assessing affinity landscapes for additional antibodies with similar epitopes would reveal how the landscape structure varies within such a group, and whether there are general features that we can extrapolate to unmeasured sequences. Finally, integrating these combinatorial libraries with saturating mutagenesis approaches would reveal how the evolvability of this lineage changes over time, and what additional mutations – such as those in BA.2, BA.4, or BA.5 – might confer further immune escape. Looking beyond the Omicron lineage, such approaches could provide more general insight into how mutations in SARS-CoV-2 may result in host-range expansion or antigenic evolution.

## Methods

### Yeast display plasmid, strains, and library production

We used the same library and strains as produced in *Rotem et al., 2018*. In brief, to generate clonal yeast strains for the Wuhan Hu-1 and Omicron BA.1 variants, we cloned the corresponding RBD gblock into a pETcon vector via Gibson Assembly. We then extracted and transformed Sanger-verified plasmids into the AWY101 yeast strain (kind gift from Dr. Eric Shusta; *Wentz and Shusta, 2007*) as described in *Gietz and Schiestl, 2007*. To produce the RBD variant library, we employed a Golden Gate combinatorial assembly strategy. We constructed full RBD sequences from five sets of dsDNA fragments of roughly equal size. Each set contains versions of the fragments that differ by the mutations included. Following bacterial transformation of this Golden Gate assembly product, we extracted and transformed the library into AWY101 yeast strain, from which we inoculated and froze a library containing obtained ~1.2 million colonies.

### High-throughput binding affinity assay (Tite-seq)

We performed Tite-seq assay as previously described (*Starr et al., 2020*; *Moulana et al., 2022*; *Adams et al., 2016*; *Phillips et al., 2021*), with two replicates for each antibody (LY-CoV016, LY-CoV555, REGN10987, and S309 [Genscript, Gene-to-Antibody service]) assay on different days, for a total of eight assays.

Briefly, we thawed yeast RBD library and the Wuhan Hu-1 and Omicron BA.1 strains by inoculating the corresponding glycerol stocks in SDCAA (6.7 g/L YNB without amino acid [VWR #90004–150], 5 g/L ammonium sulfate [Sigma-Aldrich #A4418], 2% dextrose [VWR #90000–904], 5 g/L Bacto casamino acids [VWR #223050], 1.065 g/L MES buffer [Cayman Chemical, Ann Arbor, MI, #70310], and 100 g/L ampicillin [VWR # V0339]) at 30°C for 20 hr. The cultures were then induced in SGDCAA (6.7 g/L YNB without amino acid [VWR #90004–150], 5 g/L ammonium sulfate [Sigma-Aldrich #A4418], 2% galactose [Sigma-Aldrich #G0625], 0.1% dextrose [VWR #90000–904], 5 g/L Bacto casamino acids [VWR #223050], 1.065 g/L MES buffer [Cayman Chemical, Ann Arbor, MI, #70310], and 100 g/L ampicillin [VWR # V0339]) and rotated at room temperature for 16–20 hr.

Following overnight induction, we pelleted, washed (with PBS + 0.01% BSA [VWR #45001-130; GoldBio, St. Louis, MO #A-420-50]), and incubated the cultures with mAb at a range of concentrations ($10^{-6}$ to $10^{-12}$ with 0.75-log increments for CoV555, $10^{-7}$ to $10^{-12}$ with 0.5-log increments for S309, $10^{-6}$ to $10^{-12.7}$ with 0.75-log increments for REGN10987, $10^{-6}$ to $10^{-12}$ with 0.75-log increments for SB6). The yeast-antibody mixtures were incubated at room temperature for 20 hr. The cultures were then pelleted washed twice with PBSA and subsequently labeled with PE-conjugated goat anti-human IgG (1:100, Jackson ImmunoResearch Labs #109-115-098) and FITC-conjugated chicken anti-cMmyc (1:100, Immunology Consultants Laboratory Inc, Portland, OR, #CMYC-45F). The mixtures were rotated at 4°C for 45 min and then washed twice in 0.01% PBSA.

Sorting, recovery, and sequencing library preparation followed *Moulana et al., 2022*; *Phillips et al., 2021*; *Gietz and Schiestl, 2007*. In short, we sorted ~1.2 million yeast cells per concentration, gated by FSC vs. SSC (forward vs. side scatter) and then by expression (FITC) and/or binding fluorescence (PE) on a BD FACS Aria Illu. The machine was equipped with 405 nm, 440 nm, 488 nm, 561 nm, and 635 nm lasers and an 85 micron fixed nozzle. Sorted cells were then pelleted, resuspended in SDCAA, and rotated at 30°C until late-log phase (OD600=0.9–1.4). The cultures were then pelleted and stored at –20°C for at least 6 hr prior to extraction using Zymo Yeast Plasmid Miniprep II (Zymo Research # D2004), following the manufacturer's protocol. The sequencing amplicon libraries were then prepared by a two-step PCR as previously described (*Moulana et al., 2022*; *Phillips et al., 2021*; *Nguyen Ba et al., 2019*). In brief, we added to the amplicon unique molecular identifies (UMIs), inline indices, and partial Illumina adapters through a seven-cycle PCR which amplifies the RBD sequence in the plasmid. We then used the cleaned product from the first PCR in the second PCR to append Illumina i5 and i7 indices accordingly (see https://github.com/desai-lab/compensatory_epistasis_omicron/tree/main/Supplementary_Files for primer sequences). The products were then cleaned using 0.85× Aline beads, verified using 1% agarose gel, quantified on Spectramax i3, pooled, and verified on Tapestation 5000HS and 1000HS. Final library was quantitated by Qubit fluorometer and sequenced on Illumina Novaseq SP, supplemented with 10% PhiX.

## Sequence data processing

Following *Moulana et al., 2022*, we processed raw demultiplexed sequencing reads to identify and extract the indexes and mutational sites. Briefly, for each antibody, we utilized a snakemake pipeline (https://github.com/desai-lab/omicron_ab_landscape; *Moulana, 2022*) to parse through all fastq files and group the reads according to inline indices, UMIs, and sequence reads. We accepted sequences based on criteria previously determined (10% bp mismatches) and converted accepted sequences into binary genotypes ('0' for Wuhan Hu-1 allele or '1' for Omicron BA.1 allele at each mutation position). Reads containing errors at mutation sites were removed. Finally, the pipeline collated genotype counts based on distinct UMIs from all samples into a single table.

We fit the binding dissociation constants $K_{D,app}$ for each genotype as previously described (*Nguyen Ba et al., 2019*). Briefly, using sequencing and flow cytometry data, we calculated the mean log-fluorescence of each genotype $s$ at each concentration $c$, as follows:

$$\bar{F}_{s,c} = \sum_b F_{b,c}\, p_{b,s|c}$$

where, $F_{b,c}$ is the mean log-fluorescence of bin $b$ at concentration $c$, and $p_{b,s|c}$ is the inferred proportion of cells from genotype $s$ that are sorted into bin $b$ at concentration $c$, which is estimated from the read counts as:

$$p_{b,s|c} = \frac{\frac{R_{b,s,c}}{\sum_s R_{b,s,c}} C_{b,c}}{\sum_b \left( \frac{R_{b,s,c}}{\sum_s R_{b,s,c}} C_{b,c} \right)}$$

Here, $R_{b,s,c}$ represents the number of reads from genotype $s$ that are found in bin $b$ at concentration $c$, and $C_{b,c}$ refers to the number of cells sorted into bin $b$ at concentration $c$.

We then computed the uncertainty for the mean log-fluorescence:

$$\delta \bar{F}_{s,c} = \sqrt{\sum_b \left( \delta F_{b,c}^2\, p_{b,s|c}^2 + F_{b,c}^2\, \delta p_{b,s|c}^2 \right)}$$

where, $\delta F_{b,c}$ is the spread of the log fluorescence of cells sorted into bin $b$ at concentration $c$. The error in $p_{b,s|c}$ emerges from the sampling error, which can be approximated as a Poisson process, such that:

$$\delta p_{b,s|c} = \frac{p_{b,s|c}}{\sqrt{R_{b,s,c}}}$$

Finally, we inferred the binding dissociation constant ($K_{D,s}$) for each variant by fitting the logarithm of Hill function to the mean log-fluorescence $\bar{F}_{s,c}$ , as a function of concentrations $c$:

$$\bar{F}_{s,c} = log_{10}\left( \frac{c}{c+K_{D,s}} A_s + B_s \right)$$

where, $A_s$ is the increase in fluorescence at antibody saturation, and $B_s$ is the background fluorescence level. The fit was performed using the *curve_fit* function in the Python package *scipy.optimize*. Across all genotypes, we imposed bounds on the values of $A_s$ to be $10^2$–$10^6$, $B_s$ to be $1$–$10^5$, and $K_{D,s}$ to be $10^{-14}$–$10^{-5}$. We then averaged the inferred $K_{D,s}$ values across the two replicates for each antibody after removing values with poor fit ($r^2 < 0.8$ or SE>1). Variants were defined as non-binders if the difference between the maximum and the minimum of their estimated log-fluorescence over all concentrations was lower than 1 (in log-fluorescence units). This value was set by measuring the distribution for known non-binders (see *Figure 1—figure supplement 1*).

## Isogenic measurements for validation

We validated our high-throughput binding affinity method by measuring the binding affinities for the Wuhan Hu-1 and Omicron BA.1 RBD variants. For each isogenic titration curve, we followed the same labeling strategy as in Tite-seq, titrating each antibody at concentrations ranging from $10^{-12}$-$10^{-7}$ M (with increments of 0.5 for the first replicate and 1 for the second one) for isogenic yeast strains that display only the sequence of interest. The mean log fluorescence was measured using a BD LSR Fortessa cell analyzer. We directly computed the mean and variances of these distributions for each concentration and used them to infer the value of $K_{D,app}$ using the formula shown above.

## Decision trees on loss-of-binding mutations

To summarize mutations that drive the loss of binding (escape) for each antibody, we constructed a decision tree using package rpart in R (*Therneau et al., 2013*) with its default parameters. In brief, for each antibody (except for S309 where every sequence binds the antibody), we first categorized each genotype into a binary parameter with values 'binding' or 'non-binding'. Then, the function rpart splits the tree based on any one of the 15 mutations by minimizing the Gini impurity for the binding parameter. The method continues to partition the tree if the cost complexity parameter of the split does not drop below 0.01. This parameter is the sum of all misclassifications (binding vs. non-binding) at every terminal node (analogous to residual sum of squares in regression), added by the product between the number of splits (analogous to degree of freedom) and a penalty term inferred through cross-validation performed by the rpart algorithm. The tree is then presented in *Figure 2* using 'ggparty' package (*Borkovec, 2019*).

## Epistasis analysis

We used a linear model where the effects of combinations of mutations sum to the phenotype of a sequence. The logarithm of the binding affinity $\log K_{D,app}$ is proportional to change in free energy. Thus, without epistatic interactions, the effects of mutations are expected to combine additively (*Wells, 1990*; *Olson et al., 2014*). We describe here our analysis of epistatic effects that lead to departures from this additive expectation.

Though we could naively infer all $2^{15}$ epistatic coefficients (corresponding to each subset of mutations, including all possible orders of epistasis) since we have measured binding affinities for all possible combinations of the 15 RBD mutations. However, this approach is inherently unstable: such inference will tend to identify spurious and insignificant higher-order epistatic terms to compensate for measurement errors. To avoid this problem, we truncated our model at an optimal order. That is, we neglected all epistasis terms involving more than a certain number of mutations, as is common in other analyses of epistasis (*Moulana et al., 2022*; *Phillips et al., 2021*; *Otwinowski et al., 2018*). To determine which order is optimal, we used a 10-fold cross-validation strategy by training each model on 90% of the dataset and examining its performance on the remaining 10%, as shown in *Figure 3A*.

Some phenotypic variables $\log K_{D,app}$ are unavailable in our dataset due to the upper limit of the assay concentration: we are unable to precisely infer $K_{D,app}$ for the low-affinity (or non-binding) variants, particularly when the true $-\log K_{D,app} < 6$ (the highest concentration used). To address this issue, we augmented our linear model with a lower boundary, following a Tobit left-censored model (*Tobin, 1958*). In this model, the sampling probability of $-\log K_{D,app} < 6$ is modeled using a cumulative distribution which contributes to the maximum-likelihood. Thus, the full $K$-order model can be written as:

$$-\log K^*_{D,s} = \beta_0 + \sum_{i=1}^{K} \sum_{c \in C_i} \beta_c x_{c,s} + \epsilon_s$$

where $C_i$ contains all $\binom{L}{i}$ combinations of size $i$ of the mutations and $x_{c,s}$ equal to 1 if the sequence $s$ contains all the mutations in $c$ and to 0 otherwise. Here, $-\log K_{D,s} = -\log K^*_{D,s}$ if $-\log K^*_{D,s} > 6$ and $-\log K_{D,s} = 6$ if $-\log K^*_{D,s} \leq 6$. Then, following the Tobit model approach, we compute the likelihood function to infer coefficient parameters $\beta_{MLE}$, given by:

$$\mathcal{L}(\beta, \sigma) = \prod_{j=1}^{N} \left( \frac{1}{\sigma} \varphi \left( \frac{y_j - \left( \beta_0 + \sum_{i=1}^{K} \sum_{c \in C_i} \beta_c x_{c,s} \right)}{\sigma} \right) \right)^{I(y_j)} \left( 1 - \Phi \left( \frac{\left( \beta_0 + \sum_{i=1}^{K} \sum_{c \in C_i} \beta_c x_{c,s} \right) - 6}{\sigma} \right) \right)^{1 - I(y_j)}$$

where, $y_j = -\log K_{D,app,j}$, and $\varphi$ and $\Phi$ denote the standard normal cumulative distribution function and probability density function, respectively. Moreover, note that $I(y) = \begin{cases} 0 \; if \; y \leq 6 \\ 1 \; otherwise \end{cases}$. This optimization problem would include coefficients that are associated with the loss-of-binding phenotypes. Consequently, by the model, these coefficients do not have lower bounds, and the optimization would have resulted in deflated coefficients offset by inflated higher-order coefficients, or vice-versa. To resolve this problem, we add a lasso regularization term in the form of $\epsilon \sum |\beta_c|$ to the likelihood, with $\epsilon = 0.01$. This term is small enough to reduce the magnitude of constrained coefficients but act as

intended on the non-constrained ones. In particular, we do not use the correction to make any strong assumption on the sparsity of the epistatic coefficients but to reduce instability caused by the Tobit model inference. To maximize the log-likelihood function, which is a concave function, we used the optimize module in the scipy package, with the BFGS (Broyden–Fletcher–Goldfarb–Shanno) gradient-descent method.

Although we have focused our analyses on idiosyncratic epistasis, we could also consider models of global epistasis. The significance of global epistasis in the context of protein binding is still disputed. However, global epistasis, as described by *Sailer and Harms, 2017*, might arise from changes in conformation or stability, leading the measured phenotypes to be a non-linear function of a simpler underlying linear phenotype. We have made use of this concepts and inferred global epistatic terms in previous other recent related work (*Sailer and Harms, 2017*). However, in this study, due to potential confounding effects arising from the non-binding variants (and corresponding lower boundary of detection), we chose not to perform such inference on this dataset. Moreover, we do not have any evidence that there are any changes in conformation or stability of the protein that could potentially lead to global epistatic effects in our data.

## Structural analysis

We used the reference structure of a 2.79 Å cryo-electron microscopy structure of Omicron BA.1 complexed with ACE2 (PDB ID: 7WPB). The contact surface area is determined by using ChimeraX (*Pettersen et al., 2021*) to measure the buried surface area between ACE2 and each mutated residue in the RBD (*measure buriedarea* function, default probe radius of 1.4 Å), whereas distance between α-carbons is measured using PyMol (*Schrodinger LLC, 2015*).

## Force directed layout

The high-dimensional binding affinity landscape can be projected in two dimensions with a force-directed graph layout approach (see https://desai-lab.github.io/wuhan_to_omicron/). Each node corresponds to each sequence in the library, connected by edges to a neighbor that differs in one single site. For each antibody, an edge between two sequences $s$ and $t$ is given the weight:

$$w_{s,t} = \frac{1}{0.01 + |\log K_{D,s} - \log K_{D,t}|}$$

Additionally, we also constructed a different layout that includes affinities to all antibodies, where the weight between two sequences depends on the sum over the antibodies of the difference between their affinities:

$$w_{s,t} = \frac{1}{0.01 + \sum_{a \in A} |\log K_{D,a,s} - \log K_{D,a,t}|}$$

where, $A$ is the set of antibodies we used. In a force-directed representation, the edges pull together the nodes they are attached to proportional to the weight given to each edge. In our scenario, this means that nodes with a similar genotype (a few mutations apart) and a similar phenotype (binding affinity or total binding affinity) will be close to each other in two dimensions.

Importantly this is not a 'landscape' representation: the distance between two points is unrelated to how easy it is to reach one genotype from another in a particular selection model. Practically, after assigning all edge weights, we use the layout function *layout_drl* from the Python package *iGraph*, with default settings, to obtain the layout coordinates for each variant.

## Genomic data

To analyze SARS-CoV-2 phylogeny, we used all complete RBD sequences from all SARS-CoV-2 genomes deposited in the Global Initiative on Sharing All Influenza Data (GISAID) repository (*Khare et al., 2021*; *Elbe and Buckland-Merrett, 2017*; *Shu and McCauley, 2017*) with the GISAID Audacity global phylogeny (EPI_SET ID: EPI_SET_20220615uq, available on GISAID up to June 15, 2022, and accessible at https://doi.org/10.55876/gis8.220615uq). We pruned the tree to remove all sequences with RBD not matching any of the possible intermediates between Wuhan Hu-1 and Omicron BA.1 and analyzed this tree using the python toolkit ete3 (*Huerta-Cepas et al., 2016*). We measured the frequency of each mutation by counting how many times it emerges in the tree, normalized by the

total occurrences of other mutations. For frequency with Q498R and N501Y, we counted the occurrence of each mutation only on branches that already contains Q498R and N501Y and normalized similarly.

## Statistical analyses and visualization

All data processing and statistical analyses were performed using R v4.1.0 (*R Development Core Team, 2017*) and python 3.10.0 (*Van Rossum and Drake, 2009*). All figures were generated using ggplot2 (*Wickham, 2016*) and matplotlib (*Hunter, 2007*).

## Materials and correspondence

## Data and code availability statement

Raw sequencing reads have been deposited in the NCBI BioProject database under accession number PRJNA877045. All associated metadata are available at https://github.com/desai-lab/omicron_ab_landscape, (copy archived at swh:1:rev:9ab630decfa835b2551430ed693796ef366b1aff; *Moulana, 2022*).

## Acknowledgements

We thank Zach Niziolek for flow cytometry assistance and all members of the Desai lab and Serafina Nieves for helpful discussions. TD acknowledges support from the Human Frontier Science Program Postdoctoral Fellowship, AMP acknowledges support from the Howard Hughes Medical Institute Hanna H Gray Postdoctoral Fellowship, JC acknowledges support from the National Science Foundation Graduate Research Fellowship, and MMD acknowledges support from the NSF-Simons Center for Mathematical and Statistical Analysis of Biology at Harvard University, supported by NSF grant no. DMS-1764269, and the Harvard FAS Quantitative Biology Initiative, grant PHY-1914916 from the NSF and grant GM104239 from the NIH. JDB acknowledges support from NIH/NIAID grant R01AI141707 and is an Investigator of the Howard Hughes Medical Institute. We gratefully acknowledge all data contributors, i.e., the Authors and their Originating laboratories responsible for obtaining the specimens, and their Submitting laboratories for generating the genetic sequence and metadata and sharing via the GISAID Initiative. Computational work was performed on the FASRC Cannon cluster supported by the FAS Division of Science Research Computing Group at Harvard University.

## Additional information

### Competing interests

Angela M Phillips, Michael M Desai: has or has recently consulted for Leyden Labs. Allison J Greaney, Tyler N Starr: is an inventor on Fred Hutch licensed patents related to viral deep mutational scanning (patent numbers WO2022146484, WO2020006494 and application number US20210147832). Jesse D Bloom: is an inventor on Fred Hutch licensed patents related to viral deep mutational scanning (patent numbers WO2022146484, WO2020006494 and application number US20210147832). JDB has or has recently consulted for Apriori Bio, Oncorus, Moderna, and Merck. The other authors declare that no competing interests exist.

### Funding

| Funder | Grant reference number | Author |
|---|---|---|
| Human Frontier Science Program | Postdoctoral Fellowship | Thomas Dupic |
| Howard Hughes Medical Institute | Hanna H. Gray Postdoctoral Fellowship | Angela M Phillips |

| Funder | Grant reference number | Author |
|---|---|---|
| National Science Foundation | Graduate Research Fellowship Program | Jeffrey Chang |
| National Science Foundation | Simons Center DMS-1764269 | Michael M Desai |
| National Science Foundation | Harvard Quantitative Biology Initiative DEB-1655960 | Michael M Desai |
| National Institutes of Health | Harvard Quantitative Biology InitiativeGM104239 | Michael M Desai |
| National Institutes of Health | NIH/NIAID R01AI141707 | Jesse D Bloom |

The funders had no role in study design, data collection and interpretation, or the decision to submit the work for publication.

## Author contributions

Alief Moulana, Thomas Dupic, Conceptualization, Resources, Data curation, Software, Formal analysis, Validation, Investigation, Visualization, Methodology, Writing – original draft, Project administration, Writing – review and editing; Angela M Phillips, Conceptualization, Resources, Data curation, Software, Formal analysis, Supervision, Validation, Investigation, Visualization, Methodology, Writing – original draft, Project administration, Writing – review and editing; Jeffrey Chang, Conceptualization, Resources, Data curation, Software, Formal analysis, Supervision, Validation, Investigation, Visualization, Methodology, Project administration, Writing – review and editing; Anne A Roffler, Investigation, Writing – review and editing; Allison J Greaney, Tyler N Starr, Conceptualization, Resources, Writing – review and editing; Jesse D Bloom, Conceptualization, Resources, Supervision, Writing – review and editing; Michael M Desai, Conceptualization, Resources, Funding acquisition, Writing – original draft, Project administration, Writing – review and editing

## Author ORCIDs

Alief Moulana http://orcid.org/0000-0002-0389-7082
Thomas Dupic http://orcid.org/0000-0003-1803-1617
Angela M Phillips http://orcid.org/0000-0002-9806-7574
Anne A Roffler http://orcid.org/0000-0001-8412-0322
Tyler N Starr http://orcid.org/0000-0001-6713-6904
Jesse D Bloom http://orcid.org/0000-0003-1267-3408
Michael M Desai http://orcid.org/0000-0002-9581-1150

## Decision letter and Author response

Decision letter https://doi.org/10.7554/eLife.83442.sa1
Author response https://doi.org/10.7554/eLife.83442.sa2

# Additional files

## Supplementary files

• Supplementary file 1. Isogenic validation of binding affinities. The $K_{D,app}$ inferred from isogenic measurements (see Methods) shown with those inferred via Tite-seq measurement. NB denotes non-binding and SDs between replicates are also shown.

• MDAR checklist

## Data availability

Raw sequencing reads have been deposited in the NCBI BioProject database under accession number PRJNA877045. All associated metadata are available at https://github.com/desai-lab/omicron_ab_landscape, (copy archived at swh:1:rev:9ab630decfa835b2551430ed693796ef366b1aff).

The following dataset was generated:

| Author(s) | Year | Dataset title | Dataset URL | Database and Identifier |
|---|---|---|---|---|
| Desai MM | 2023 | The landscape of antibody binding affinity in SARS-CoV-2 Omicron BA.1 evolution | http://www.ncbi.nlm.nih.gov/bioproject/?term=PRJNA877045 | NCBI BioProject, PRJNA877045 |

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
