## [Editor Report]

This fundamental study, dealing with antibody binding to spike protein of the omicron variant of SARS CoV2, advances our insights into antibody escape and the importance of epistasis in antibody binding. The evidence is rigorous and compelling. The work will be of great interest to investigators in the field of evolutionary biology/medicine, immunologists, and virologists.

---

## [Decision Letter]

**Decision letter after peer review:**

Thank you for submitting your article "The landscape of antibody binding affinity in SARS-CoV-2 Omicron BA.1 evolution" for consideration by eLife. Your article has been reviewed by 2 peer reviewers, and the evaluation has been overseen by a Reviewing Editor and Jos van der Meer as the Senior Editor. The reviewers have opted to remain anonymous.

Essential revisions:

As was brought forward by reviewer 2, we would like to see more of a dive into the peculiarity of the finding with respect to epistasis. More details can be found in review #2 below.

*Reviewer #2 (Recommendations for the authors):*

In addition to the comments in my public review, I wanted to emphasize more interpretation of the specific results. This project offers an incredibly impressive amount of labor, and from my lens, seems to be analyzed properly.

Like I suggested, I'd prefer to have the epistasis results couched with respect to other modern studies of epistasis in proteins (even in SARS-CoV-2). And I would the methods used to detect epistasis to be embedded in greater modern discussions of how higher-order epistasis shapes landscapes of various kinds.

All in all, a terrific study.

---

## [Author Response]

Essential revisions:As was brought forward by reviewer 2, we would like to see more of a dive into the peculiarity of the finding with respect to epistasis. More details can be found in review #2 below.

In this revision, we have added a paragraph to the main text and more information in the methods section, as requested by the reviewers. We have also corrected an error in our preprocessing pipeline, and updated figures to reflect these slight modifications.